# Inadequacy of Vitamin D Does Not Interfere with Body Weight Loss in Women of Reproductive Age after Roux-en-Y Gastric Bypass

**DOI:** 10.3390/biomedicines11010086

**Published:** 2022-12-29

**Authors:** Sabrina Cruz, Andrea Cardoso de Matos, Suelem Pereira da Cruz, Silvia Pereira, Carlos Saboya, Andrea Ramalho

**Affiliations:** 1School of Medicine, Federal University of Rio de Janeiro (UFRJ), Rio de Janeiro 21941-902, Brazil; 2Center for Research on Micronutrients (NPqM), Institute of Nutrition Josué de Castro of UFRJ, Rio de Janeiro 21941-902, Brazil; 3Nutrition Federal University Fluminence (UFF), Rio de Janeiro 21941-902, Brazil; 4Multidisciplinary Center, Bariatric and Metabolic Surgery, Rio de Janeiro 21941-902, Brazil; 5Escola Paulista de Medicina, Federal University of São Paulo (UNIFESP), São Paulo 04021-001, Brazil; 6Department of Social and Applied Nutrition, the Institute of Nutrition at UFRJ, Rio de Janeiro 21941-902, Brazil

**Keywords:** Roux-en-Y gastric bypass, bariatric surgery, weight reduction, obesity

## Abstract

Objective: To assess the influence of vitamin D on body weight loss in women who had previously undergone Roux-en-Y gastric bypass (RYGB). Methodology: This is an analytical, longitudinal and retrospective study comprising 40 women of reproductive age who had previously undergone RYGB. To investigate the influence of the serum concentrations of vitamin D on body weight reduction, the variables were analyzed in the pre-operative period (T0), in the first (T1) and in the second postoperative year (T2) and were stratified according to the BMI measured in T1 and T2. In addition, in the pre-operative period, participants were subdivided into groups based on adequacy (G1), deficiency (G2) and insufficiency (G3), according to their serum concentrations of vitamin D. Results: Although weight loss occurred in a substantial way in T1, it continued to decrease in T2 (*p* = 0.017). The women who reached normal weight within two years of surgery showed the lowest vitamin D concentrations preoperatively when compared to those who were overweight (*p* = 0.011). Women with preoperative vitamin D deficiency showed increased concentrations in the assessed times (*p* < 0.001), while the opposite (*p* = 0.001) occurred in women with adequacy. Conclusion: The study showed that inadequacy of vitamin D does not interfere with weight loss in the two-year-follow-up after RYGB and highlights that vitamin D can present a differentiated response postoperatively, to the detriment of the pre-operative period.

## 1. Introduction

Worldwide, both obesity and reduced vitamin D concentrations are considered serious public health issues [1,2]. One of the treatment alternatives for weight reduction in individuals with severe obesity is bariatric surgery, and among them, the Roux-en-Y gastric bypass (RYGB) stands out as one of the most common performed procedures [3].

In patients who had undergone bariatric surgery, inadequacy of vitamin D recurs in the pre and postoperative periods [4,5] and can affect 90% of individuals with obesity [6]. It is important to highlight that this vitamin can influence the nutritional status of calcium with its homeostasis regulated by the parathyroid hormone [7]. The high prevalence of the inadequacy of vitamin D in obese individuals can be mainly explained by volumetric dilution; decreased skin synthesis capacity; ‘sequestration’ in adipose tissue; metabolic changes in obese individuals as a result from reduction of the enzymatic activity of the 1alpha-hydrolase that increases degradation and decreases the synthesis of this nutrient [8,9,10].

During and/or after body weight reduction promoted by bariatric surgery, vitamin D metabolism can be modified and can promote inadequacy [11,12]. This scenario contributes to increased overweight/obesity [13] and is associated with excess weight reduction after 2 years of surgery [5]. In this respect, vitamin D supplementation may be related to higher weight reduction [14]; however, clinical trials have argued against such possibility [15,16].

Despite its importance, few studies have investigated the direct effect of the serum concentrations of vitamin D on body weight reduction in individuals with obesity [14]. However, studies addressed to the effects of vitamin D on weight loss and weight maintenance in individuals who had undergone bariatric surgery have not been found in the literature yet. Given this, our aim is to investigate the influence of the nutritional status of vitamin D on body weight reduction after 1 and 2 years of RYGB. We will also verify whether the serum concentrations of vitamin D in the pre-operative period can contribute to surgical success, excess weight and weight loss.

## 2. Methodology

This is an analytical, longitudinal and retrospective study comprising 40 white women of reproductive age who had previously undergone RYGB, carried out at the Multidisciplinary Center for Bariatric and Metabolic Surgery of the municipality of Rio de Janeiro from January 2011 to July 2016.

For the investigation of the influence of the serum concentrations of vitamin D3, calcium and parathyroid hormone (PTH) on body weight reduction and/or obesity, participants were stratified according to the BMI measured in the first and second years after surgery. In addition, in the pre-operative period, participants were subdivided into adequacy (G1), deficiency (G2) and insufficiency (G3) (Figure 1), according to the serum concentrations of vitamin D3.

All participants were followed up by the clinic specialized in obesity control and responsible for performing the surgery. In the event of the inadequacy of vitamin D in the preoperative period, all women received a daily intake of 1500 IU of vitamin D3, following a recommendation from the Brazilian Society of Endocrinology and Metabolism. They followed a single protocol of supplementation of daily vitamins and minerals set up by the clinic in compliance with the IOM recommendations (2009) [17], with 850 mg of calcium carbonate and 600 IU of vitamin D3. To assess the adherence to the proposed protocol of supplementation, the containers of the prescribed supplements were requested in every consultation. During these consultations, the importance of the daily use of the supplements was emphasized and educational material was given to the patients with information about the benefits resulting from supplementation.

By consulting the medical records, information about age and height in the pre-op, as well as pre-op weight and weight after 1 and 2 years of surgery, was collected. From these variables, an anthropometric assessment of the patients was performed, and the body mass index (BMI) was calculated according to the WHO (1998) cutoffs [18]; weight loss (WL), excess weight (EW), percentage of weight loss and surgical success, considering the following equations:

1. Weight loss (WL) = initial weight − current weight;

(G1) WL = preoperative weight − weight after 1 and/or 2 years of surgery (G2) WL = preoperative weight − pre-gestational weight

2. For excess weight (EW) = actual weight − ideal weight, where the ideal weight (IW) is given by IW = 53.975 + [(height − 1.524) × 53.5433].

(G1) real weight = weight after 1 and/or 2 years of surgery

(G2) real weight = pre-gestational weight

3. Surgery success: for a satisfactory result, patients should be in the following condition: BMI < 30 kg/m^2^ for patients with initial BMI < 50 kg/m^2^ or BMI < 35 kg/m^2^ for patients with initial BMI ≥ 50 kg/m^2^ [19,20,21].

4. Percentage of weight loss: excellent, corresponds to loss > 35%; good, loss between 25 and 34%; poor, loss between 15 to 24%; and for surgical failure, a ponderal loss < 15% at the end of 1 year [22].

The analysis of vitamin D was conducted in the form of 25(OH)D and the method used for its quantification was high-efficiency liquid chromatography with ultraviolet detector (HPLC-UV). The serum values obtained were compared to the cutoff points for normality proposed by Holicket et al. (2011). In this way, the serum concentrations of vitamin D were classified as deficiency (≤20 ng/mL), insufficiency (≥21 ng/mL and <29 ng/mL) or adequacy (≥30 ng/mL and <100 ng/mL).

The instrument used in data collection was pretested, comprising a form filled by a single interviewer through an interview and analysis of prenatal medical records and complemented by a nutritionist consultation. The overall study participants read and signed the Free and Informed Consent Form in compliance with the National Health Council Resolution. The study was approved by the Research Ethics Committee of the University Hospital Clementino Fraga.

In relation to the statistical analysis, the Kolmogorov–Smirnov and Shapiro–Wilk tests were used for normality verification. Quantitative data were described in measures of central tendency and dispersion, and the Mann–Whitney or Kruskal–Wallis test was used for comparison of means. To test for homogeneity of proportions between categorical variables, the Chi-square test was applied. Correlations were performed and assessed by the Spearman correlation coefficient. In the analyses, a 5% significance level was adopted. The overall statistical assessment was carried out in the SPSS statistical package for Windows version 21.0.

## 3. Results

### 3.1. Sample Characterization and Description of Anthropometric Variables

The study comprised 40 women who had previously undergone RYGB, of reproductive age, between 20 and 42 years and an average of 34.00 ± 5.62 years. BMI mean in the preoperative period was 42.85 ± 3.50 kg/m^2^, which decreased to 27.67 ± 3.60 kg/m^2^ (*p* < 0.001) in T1 and to 25.97 kg/m^2^ ± 2.75 in T2 (*p* < 0.001). When comparing T1 to T2, it was observed that mean BMI was significantly lower in T2 (*p* = 0.012).

In the preoperative period, 80.0% of the women were classified with class III obesity and 20.0% with class II obesity. In the first year after surgery, 22.5% of women reached normal weight, 52.5% overweight, 22.5% class I obesity and only 2.5% class II obesity. In the second year, the percentage of normal weight increased by 35.0%, as well as overweight reached 62.5% and only 2.5% remained in class II obesity.

In T1, 80.0% achieved surgical success that reached 97.5% in T2 (*p* = 0.613). Mean weight loss and excess weight were, respectively: T1 = 40.87 ± 7.72 kg vs. T2 = 45.55 ± 8.25 kg, with *p* = 0.017 and T1 = 14.66 ± 10.47 vs. T2 = 9.98 ± 8.40, with *p* = 0.014.

In the first year after RYGB, 55% and 40% presented percentage of WL classified as excellent and good, respectively, with only 5% poor. After 2 years, 80% had excellent weight loss, with only 17.5% good and 2.5% poor. There were no significant differences between the times (*p* = 0.296).

### 3.2. Nutritional Status of Vitamin D

In the preoperative, mean serum concentrations of vitamin D was inadequate, with 80.0% of affected women. After surgery, means and percentages of inadequacy were maintained in the two assessed times. However, it is worth highlighting that, in the postoperative period, the percentage of women affected by VDD increased by 25% in T1 and only 10% in T2 (Table 1).

### 3.3. Influence of the Nutritional Status of Vitamin D on Surgical Success, Excess Weight, Weight Loss and Percentage of Weight Loss in the Assessed Times

When women were subdivided by adequacy of vitamin D in the preoperative period, it was noted that those classified with deficiency showed an increase in the serum concentrations of this nutrient in T1 (*p* ≤ 0.001) and T2 (*p* ≤ 0.001), with a shift from deficiency to insufficiency. Conversely, women with adequacy of vitamin D in the preoperative period showed reductions in T1 (*p* = 0.001) and T2 (*p* = 0.004) (Table 2) *.

It was observed that women with VDD in the preoperative period showed the lowest BMI mean when compared to those with adequacy (*p* = 0.02) in T2 (Table 3) *. Moreover, there was a correlation between the serum concentrations of vitamin D in the preoperative period with BMI (r = 0.483 with *p* = 0.002) and EW (r = 0.324 with *p* = 0.041) 2 years after RYGB.

As regards the quality of weight loss, it was noted that 95% of women in T1 had an excellent and/or good classification regardless of the nutritional status of vitamin D in the pre-operative period (*p* = 0.301). A similar result was found in T2 with a percentage of 97.5% (*p* = 0.500).

### 3.4. Assessment of the Nutritional Status of Vitamin D

The current study notes that body weight reduction and, consequently, BMI in the first postoperative year did not relate to the serum concentrations of vitamin D (Table 4). Despite this, overweight patients showed mean PTH with significant differences between the assessed times (*p* = 0.04) in which T0 and T1 had the lowest means when compared to T2 (T0:49.83 ± 13.11 vs. T2:59.47 ± 15.80, *p* = 0.022; T1:48.47 ± 16.35 vs. T2:59.47 ± 15.80, *p* = 0.043), and patients that reached normal weight had higher percentages in the same period (T0:0% vs. T1:0% vs. T2:33.3%, *p* = 0.034). In addition, they all had calcium above normal values, regardless of the BMI at the assessed times.

Women who reached normal weight two years after RYGB showed lower mean vitamin D when compared to overweight (*p* = 0.014) and/or excess weight women (*p* = 0.011) in T0 (Table 4) *. In the 2nd year, women with normal weight had vitamin D concentrations correlated with BMI (r = −0.797, *p* = 0.001), EW (r = −0.784, *p* = 0.001) and percentage of WL (r = 0.731, *p* = 0.003) in T2.

In addition, 1 year after RYGB, overweight and/or class I obesity women had vitamin D concentrations in T0 correlated with percentage of WL (r = 0.514, *p* = 0.017; r = 0.833, *p* = 0.005, respectively) and overweight women with EW (r = −0.591, *p* = 0.005).

## 4. Discussion

The literature has been consistent regarding the association between the serum concentrations of vitamin D and obesity [13,23], and it has pointed out that 1 kg/m^2^ increase in BMI can cause a decline of 1.15% of this nutrient [24]. In patients in the postoperative period, VDD can affect 50–80% of individuals [25]. In addition, Costa et al. (2014) [26] have reported that this deficiency has affected 60.41% of the participants and that no correlation with solar exposure has been found.

In the present study, the inadequacy of vitamin D in the preoperative period affected 80% of women, with half of these classified with deficiency. However, despite weight reduction in the assessed times, no significant differences were found regarding their serum concentrations.

After RYGB, patients can reduce more than 35% of their initial weight and 62–75% of excess weight [27] in the first 12 months after bariatric surgery. This period is known as the period of the highest catabolism resulting from fast body weight reduction [28]. However, the regain of this adipose tissue can occur in the second year after bariatric surgery [29,30].

In fact, this surgical procedure contributes to body weight reduction, and in the current study, we have noted an expressive decrease in the first postoperative year. However, it is worth pointing out the continuity of weight reduction when we compare T1 to T2, since in this last period there were the highest weight loss means (*p* = 0.017), the lowest BMI means (*p* = 0.012) and the lowest EW means (*p* = 0.014), as well as the highest percentage of WL classified as excellent, and almost all women achieved surgical success.

It has been recognized that the adipose tissue is the largest endocrine organ that can regulate and/or be regulated by vitamin D [31,32]. The complex biochemical interactions between this nutrient and the in vitro adipose tissue raise the question whether the inadequacy of vitamin D, per se, can contribute to obesity or inhibits body weight reduction [13].

In this sense, the current study suggests that the inadequacy of vitamin D may not interfere with body weight reduction, since those that reached normal weight after two years of RYGB showed the lowest vitamin D concentrations in the preoperative period when compared to those overweight (*p* = 0.014) and/or with excess weight (*p* = 0.011). Furthermore, more than 95% of women presented with a percentage of WL classified as excellent and/or good in the pre-operative period [T1:95% with *p* = 0.301; T2:97.5% with *p* = 0.500], regardless of the nutritional status of vitamin D in the preoperative period.

Another possible factor that can contribute to the prevention of weight gain would be related to the adequacy of vitamin D concentrations in individuals with deficiency [33]. However, the present study has found opposite results. When we subdivided the patients according to the serum concentrations of vitamin D in the pre-operative period (adequacy, deficiency and insufficiency) and compared them according to the BMI means measured 1 year after RYGB, results were similar.

In addition, it is worth pointing out that women with VDD in the preoperative period showed increased concentrations at the assessed times (*p* < 0.001), while in those with adequacy of VD, the opposite occurred (*p* = 0.001). In this context, it can be suggested that, in the process of weight reduction, vitamin D, which is a liposoluble nutrient stored in adipocytes, [10] may have greater release of adipose tissue stocks in case of DVD in the preoperative period, which may contribute to an increase in its serum concentrations. Thus, to our knowledge, this is the first study that reports that the nutritional status of vitamin D can present differentiated response in the postoperative period. Such findings are important and deserve complementary investigations, in view of limitations such as the small sample size.

The results presented here raise an important issue regarding the nutritional status of vitamin D and body weight reduction, under the conditions in which this study was carried out. No less important are our findings in this study that show that vitamin D can have a differentiated response in the postoperative period, regardless of the serum concentrations of that nutrient in the preoperative period.

## 5. Conclusions

This study has noted that the inadequacy of the nutritional status of vitamin D does not interfere with body weight reduction in the 2 years after RYGB and suggests that vitamin D may display a differentiated response in the postoperative period in contrast to the preoperative period.

## Figures and Tables

**Figure 1 biomedicines-11-00086-f001:**
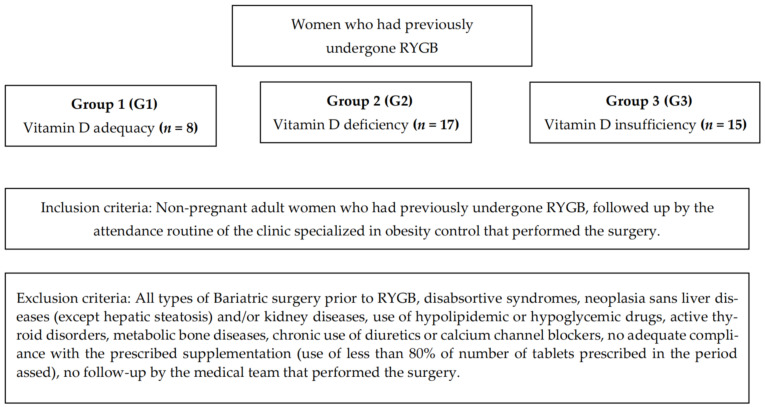
Description of the subdivision of groups and inclusion/exclusion criteria. G1: women with adequacy in the pre-operative nutritional status of Vitamin D; G2: women with deficiency in the pre-operative nutritional status of Vitamin D; G3: women with insufficiency in the pre-operative nutritional status of Vitamin D.

**Table 1 biomedicines-11-00086-t001:** Mean serum concentration and percentage of inadequacy of vitamin D3, calcium and PTH in the assessed times.

Vitamin D3 (ng/mL)	T0 (*n* = 40)	T1 (*n* = 40)	T2 (*n* = 40)	*p*-Value
Mean/Standard deviation	22.87 ± 9.70	24.77 ± 6.96	23.75 ± 6.22	0.422
% of Adequacy	20	15	15	0.239
% of Inadequacy	80	85	85	0.239
% of Insufficiency	37.5	62.5	47.5	0.787
% of Deficiency	42.5	22.5	37.5	0.767

The Kruskal–Wallis test for continuous variables or the Chi-square test for categorical variables (* *p* < 0.05); T0 = preoperative; T1 = 1 year after RYGB; T2 = 2 years after RYGB.

**Table 2 biomedicines-11-00086-t002:** Mean serum concentration of vitamin D3 in the assessed times, according to the classification of the nutritional status of vitamin D3 in the preoperative period.

Vitamin D3 in the Preoperative (ng/mL)	T0	T1	T2	*p*-Value
G1	36.12 ± 9.84	23.37 ± 3.06	23.65 ± 6.65	0.001 *
G2	14.58 ± 3.85	25.17 ± 7.86	23.47± 6.56	<0.001 *
G3	25.20 ± 2.73	25.06 ± 7.63	24.20 ± 6.00	0.965
*p*-value	<0.001 *	0.667	0.730	

The Kruskal–Wallis test for continuous variables (* *p* < 0.05); T0 = preoperative period; T1 = 1 year after RYGB; T2 = 2 years after RYGB.

**Table 3 biomedicines-11-00086-t003:** Mean anthropometric variables in the assessed times, according to the adequacy of vitamin D3 nutritional status in the preoperative period.

Anthropometric Variables	Vitamin D3 in the Preoperative Period (ng/mL)							
BMI (kg/m^2^)	G1	T1	G3	*p*-value	G1	T2	G3	*p*-value
		G2				G2		
	28.92 ± 4.82	27.24 ± 3.00	27.49 ± 3.61	0.746	26.84 ± 1.54	24.97 ± 1.93	26.63 ± 3.68	0.045 *
Surgical success	62.5%	88.2%	80%	0.324	100%	100%	93.3%	
								0.425
Excess weight (kg)	16.98 ± 13.36	14.23 ± 9.02	13.92 ± 10.92	0.868	11.41 ± 4.72	7.82 ±6.02	11.67 ± 11.59	0.248
Weight loss (kg)	44.63 ± 5.36	38.92 ± 7.97	41.07 ± 8.15	0.208	50.21± 11.21	45.34 ± 6.11	43.32 ± 8.23	0.353
% of weight loss	37.33 ± 5.43	34.13 ± 7.35	35.84 ± 6.56	0.599	41.23 ± 4.96	39.60 ± 4.80	37.81 ± 6.27	0.399

The Kruskal–Wallis test was used for continuous variables (* *p* < 0.05); T0 = preoperative period; T1 = 1 year after RYGB; T2 = 2 years after RYGB.

**Table 4 biomedicines-11-00086-t004:** Mean serum concentrations and percentage of inadequacy of vitamin D3, calcium and PTH, according to body mass index 1 and 2 years after Roux-en-Y gastric bypass.

Mean Vitamin D3 (ng/mL)				% of Inadequacy/Deficiency of Vitamin D3 (ng/mL)				
BMI after 1 Year	T0	T1	T2	*p*-value	T0	T1	T2	*p*-Value
Normal weight	22.11 ± 5.37	25.88 ± 6.03	26.88 ± 5.71	0.519	88.8/44.4	88.9/22.2	66.7/33.3	0.502/0.595
Overweight	21.71 ± 12.28	23.28 ± 7.01	21.71 ± 6.05	0.556	85.7/52.4	85.7/23.8	94.4/47.6	0.225/0.131
Obesity I	25.55 ± 5.87	27.11 ± 7.99	25.33 ± 6.30	0.947	66.7/22.2	77.8/22.2	77.8/22.2	0.980/1
Obesity II	30	25	24	0.368	-	4	5.3	-
*p*-value	0.205	0.655	0.143		0.304	0.981	0.361	
Normal weight	22.11 ± 5.37	25.88 ± 6.03	26.88 ± 5.71	0.519	88.8/44.4	88.9/22.2	66.7/33.3	0.502/0.595
Excess weight	23.09 ± 10.7	24.45 ± 7.27	22.83 ± 6.14	0.067	77.4/41.9	83.9/22.6	90.3/38.7	0.721/0.228
*p*-value	0.948	0.505	0.067		0.734	0.928	0.207	
BMI after 2 years	T0	T1	T2	*p*-value	T0	T1	T2	*p*-value
Normal weight	18.21 ± 5.84	23.35 ± 6.61	22.57 ± 6.68	0.100	92.6/64.3	85.7/28.6	92.9/50	0.443/0.163
Overweight	25.32 ± 10.72	25.40 ± 7.26	24.16 ± 6.00	0.898	72.0/32.0	84/20	84/32.0	0.487/0.551
Obesity II	27.00	29.00	30.00	1	100/0	100/0	0/0	0.386/0.223
*p*-value	0.039 *	0.390	0.304	0.100	0.186	0.911	0.125	0.443/0.163
Normal weight	18.21 ± 5.84	23.35 ± 6.61	22.57 ± 6.68		92.6/64.3	85.7/28.6	92.9/50	
Excess weight	25.38 ± 10.51	25.53 ± 7.15	24.38 ± 5.99	0.363	73.1/30.8	84.6/19.2	80.8/30.8	0.670/0.439
*p*-value	0.011 *	0.296	0.363		0.098	0.795	0.390	

The Mann–Whitney test or the Kruskal–Wallis test was used for continuous variables and the Chi-square test for categorical variables (* *p* < 0.05); T0 = preoperative period; T1 = 1 year after RYGB; T2 = 2 years after RYGB.

## Data Availability

Not applicable.

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
