# Peer review of "Inadequacy of Vitamin D Does Not Interfere with Body Weight Loss in Women of Reproductive Age after Roux-en-Y Gastric Bypass"

_biomedicines, 2022, doi:10.3390/biomedicines11010086_

Round 1

Reviewer 1 Report

This is a good study on the relationship between vitamin D levels and weight loss in patients after bariatric surgery.

The authors should compare their findings with those of other authors.  

The authors should try to show the significance of their findings.

Author Response

Response to Reviewer 1 Comments

Point 1: This is a good study on the relationship between vitamin D levels and weight loss in patients after bariatric surgery.The authors should compare their findings with those of other authors.  The authors should try to show the significance of their findings.

Response 1: In the discussion, the follow-up of individuals with vitamin D deficiency in the preoperative period was highlighted, as they had an increase in their concentrations in the two times evaluated. In this context, it can be suggested that in the process of weight reduction vitamin D, which is a liposoluble nutrient stored in adipocytes, may have greater release of adipose tissue stocks in case of DVD in the preoperative period, and may contribute to an increase in its serum concentrations.

Reviewer 2 Report

The aim is to investigate the influence of the nutritional status of vitamin D on body weight reduction after 1 and 2 years of RYGB. The authors will also verify whether the serum concentrations of vitamin D in the pre-operative period can contribute to surgical success, excess weight and weight loss. This study has noted that the inadequacy of the nutritional status of vitamin D does not interfere with body weight reduction in 2 years after RYGB and suggests that vitamin D may display a differentiated response in the postoperative period, to the detriment of the preoperative period.

The introduction is well written , with adequate bibliographic references and stating the hypothesis of the study

The methodology is complete, widely described, which would allow the study to be carried out by another research group. Concerns: ·      The absence of a control group could a limitation to the study ·      The statistical power of the study to determine the sample size has not been calculated.   Results are clearly described. A graphic representation of the tables could help to understand the results

The discussion is correct, adapting to the results obtained. However, the study  reports that the nutritional status of vitamin D can present differentiated response in the postoperative period. Some hypothesis must be suggested

Author Response

Response to Reviewer 2 Comments

Point 1: The aim is to investigate the influence of the nutritional status of vitamin D on body weight reduction after 1 and 2 years of RYGB. The authors will also verify whether the serum concentrations of vitamin D in the pre-operative period can contribute to surgical success, excess weight and weight loss. This study has noted that the inadequacy of the nutritional status of vitamin D does not interfere with body weight reduction in 2 years after RYGB and suggests that vitamin D may display a differentiated response in the postoperative period, to the detriment of the preoperative period.

The introduction is well written , with adequate bibliographic references and stating the hypothesis of the study

The methodology is complete, widely described, which would allow the study to be carried out by another research group. Concerns: The absence of a control group could a limitation to the study. The statistical power of the study to determine the sample size has not been calculated.   Results are clearly described. A graphic representation of the tables could help to understand the results

The discussion is correct, adapting to the results obtained. However, the study  reports that the nutritional status of vitamin D can present differentiated response in the postoperative period. Some hypothesis must be suggested

Response 1: The absence of a control group could a limitation to the study was added as a limitation of the study. This is a study with non-parametric statistics, due to the "n" of the present study.

Response 2: A possible hypothesis has been added. In this context, it can be suggested that in the process of weight reduction vitamin D, which is a liposoluble nutrient stored in adipocytes, may have greater release of adipose tissue stocks in case of DVD in the preoperative period, and may contribute to an increase in its serum concentrations.

Round 2

Reviewer 1 Report

 The authors have made the necessary amendments.  

Author Response

The authors have made the necessary amendments.  Comment: the scientific article is being sent with some tweaks.

Reviewer 2 Report

The authors do not respond to any of the questions raised. The statistical power of the study based on the sample size is not indicated. There is no graphical representation of the results to facilitate reading. In the discussion a hypothesis is proposed but without bibliographical references

Author Response

Response to Reviewer 2 Comments

Reviewer 2: The authors do not respond to any of the questions raised. The statistical power of the study based on the sample size is not indicated. There is no graphical representation of the results to facilitate reading. In the discussion a hypothesis is proposed but without bibliographical references.

Comments 1:

We forward some clarification of the relevant points forwarded by the reviewer: In the last paragraph of the methodology we indicate the statistical analyzes used, which in turn support the statistical power presented in our study. Thus, we highlight the use of the following tests: Kolmogorov-Smirnov and Shapiro-Wilk, which were used to verify normality. Quantitative data described in measures of central tendency and dispersion and the Mann-Whitney or Kruskal-Wallis test, used to compare means. To test the homogeneity of proportions between categorical variables, we applied the chi-square test. Correlations were performed and evaluated using Spearman's correlation coefficient. In the analyses, we adopted a significance level of 5%. We emphasize that all statistical analyzes were performed in SPSS for Windows version 21.0.

Comments 2:

In fact, most of the time, the graphical representation makes it much easier to read and understand the results. However, as the present study presents different variables for three times and in three different groups, the graphic representation that we elaborated did not help much, we even think that it made the interpretation of the results a little more confusing. For this reason, we chose to enter the means and standard deviation in table form. So, we find that in this way it became a little easier the understanding and accuracy of the absolute values found through the inclusion of the p-value of the statistical analysis carried out.

Comments 3: The bibliographic reference has been added.

Round 3

Reviewer 2 Report

The questions have been answered